# Moisture-Assisted Formation of High-Quality Silver Nanowire Transparent Conductive Films with Low Junction Resistance

**Lipeng Zhou** **, Yuehui Hu \*, Hao Gao, Youliang Gao, Wenjun Zhu, Lilin Zhan, Huiwen Liu, Yichuan Chen, Keyan Hu, Pengfei Wang, Di Wang, Fang Hu, Ke Zhou, Wei Liu and Ning Chang**

School of Mechanical and Electronic Engineering, Jingdezhen University of Ceramics Institute, Jingdezhen 333001, China; zlp10211021@163.com (L.Z.); ghao2014@126.com (H.G.); gaocsu@163.com (Y.G.); zwj2730038@126.com (W.Z.); scf970429@outlook.com (L.Z.); Liuhuiwen002@126.com (H.L.); lezhi2005@163.com (Y.C.); hukeyan123@126.com (K.H.); wpf2469@163.com (P.W.); z17802581592@yeah.net (D.W.); Hf1369790258@126.com (F.H.); kekekezhou@163.com (K.Z.); Tiantian1332@yeah.net (W.L.); NingHou1220@163.com (N.C.)
\* Correspondence: huyuehui@jci.edu.cn; Tel.: +86-13979803899

**Abstract:** Silver nanowire (AgNWs) transparent conductive film (TCF) is considered to be the most favorable material to replace indium tin oxide (ITO) as the next-generation transparent conductive film. However, the disadvantages of AgNWs, such as easy oxidation and high wire-wire junction resistance, dramatically limit its commercial application. In this paper, moisture treatment was adopted, and water was dripped on the surface of AgNWs film or breathed on the surface so that the surface was covered with a layer of water vapor. The morphology of silver nanowire mesh nodes is complex, and the curvature is large. According to the capillary condensation theory, water molecules preferentially condense near the geometric surface with significant curvature. The capillary force is generated, making the wire-wire junction of AgNWs mesh bond tightly, resulting in good ohmic contact. The experimental results show that AgNWs-TCF treated by moisture has better conductivity, with an average sheet resistance of 20 $\Omega$/sq and more uniform electrical properties. The bending test and adhesion test showed that AgNWs-TCF treated by moisture still exhibited good mechanical bending resistance and environmental stability.

**Keywords:** silver nanowires; transparent conductive film; moisture treatment; capillary condensation; optoelectronic properties

## 1. Introduction

With the rapid development of electronic science and technology, transparent conductive oxide (TCF) is widely used in touch panels, solar cells [1], organic light emitting diode [2], wearable devices [3], window defrosting, or thermal insulation window [4], etc. At present, indium tin oxide (ITO) transparent conductive film has outstanding photoelectric properties compared with other materials [5,6] and, is the most widely used material in commerce, with a market share as high as 95%. However, the scarcity of indium resources, high deposition temperature, expensive preparation cost, and essentially inherent brittleness largely limits its application in flexible electronics devices [7–9]. Therefore, it was crucial to research and find a new generation of materials that can replace ITO as transparent conductive film [10,11]. Silver nanowire transparent conductive film (AgNWs-TCF) is considered the best substitute for ITO because of its excellent optical transparency, excellent conductivity, and good mechanical flexibility [12–14].

However, the high wire-wire junction resistance and poor adhesion to the substrate of AgNWs-TCF restrict its commercial application [15,16]. There are two reasons for the poor resistance of AgNWs network nodes. First, polyvinylpyrrolidone (PVP) will inevitably synthesize AgNWs from polyols, which have insulating properties [17]. Second, The wire–wire connection in AgNWs network nodes is not tight, resulting in poor ohmic contact [18]. At

present, scientific and technical workers mainly adopt the sandwich structure of metal oxide coating AgNWs to make the wire–wire junction in silver nanowires tightly bonded [19], graphene material modification to increase conductive channels [20], ultraviolet lamp irradiation to produce the wire–wire junction in AgNWs welding [21], thermal annealing to melt PVP, laser welding nanowire nodes to improve mechanical properties [22], CNT modification to enhance the mechanical flexibility and photoelectric properties [23], and current-induced Joule thermal welding of silver nanowires [24], conducting polymer PE-DOT:PSS ink-assisted joining [25], etc., to solve the problem of high wire-wire junction resistance of AgNWs network and have made good progress [26–28].

Nevertheless, considering the factors affecting industrial development, such as process and cost [29], we developed a moisture treatment method to realize the tight wire–wire connection in AgNWs network nodes and produce good ohmic contact. This water was applied to the AgNWs network prepared by the spin-coating method, and water molecules were adsorbed on the curved surface of AgNWs net nodes. Due to capillary force generated by rapid evaporation of water molecules [30], AgNWs network nodes were tightly combined, the number of conductive channels increased, and the conductivity of AgNWs-TCF improved. This moisture treatment method is simple, feasible, and has industrialization potential.

## 2. Materials and Methods

### 2.1. Materials

Silver nanowires (with a diameter of 50 nm and an average length of 25–30 μm, from Nanjing Xianfeng Nanomaterials Technology Co., Ltd., Nanjing, China) were dispersed in isopropanol solution, and the concentration in the solution was 5 mg/mL. Glass (area = 15 × 15 mm$^2$) and PET substrate (area = 15 × 15 mm$^2$) purchased from Gulo Glass Co., Ltd., Luoyang, China.

### 2.2. Formation of AgNWs Transparent Conductive Films

In the first stage, 0.5 mL AgNWss solution was spin-coated and deposited on the substrate by KW-4A spin coater. To begin with, the spinning speed was set to 500 r/min for a duration of 10 s in low gear, then 1100 r/min for a duration of 30 s in high gear. The wet film of AgNWs was heated and dried at 100 °C for 10 min to evaporate solvent residue, and AgNWs-TCF was obtained after cooling to room temperature. In the second stage, a small amount of deionized water dripped on the surface of AgNWs-TCF prepared in the first stage. After 1 min, the second layer of AgNWs-TCF was deposited by repeating the spin coating process in the first stage, and finally, the moisture treatment AgNWs-TCF was obtained. The schematic diagram of the experimental process is shown in Figure 1.

The reason for the selection of heating temperature is that the boiling point of water is 100 °C and will not cause thermal welding of the nanowires.

Ordinary water or liquid has a certain amount of cations, while the solution of AgNWs has the characteristics of polyanion. The combination products of cations in liquid and anions adsorb on the surface of AgNWs will affect the light transmittance of AgNWs-TCFs, which is why deionized water is chosen to treat AgNWs.

The AgNWs-TCFs without water moisture treatment are also spin-coated with two layers of AgNWs solution. The preparation process of AgNWs-TCFs is the same as that of AgNWs-TCFs treated by water moisture, except that the second stage of water moisture treatment is not required.

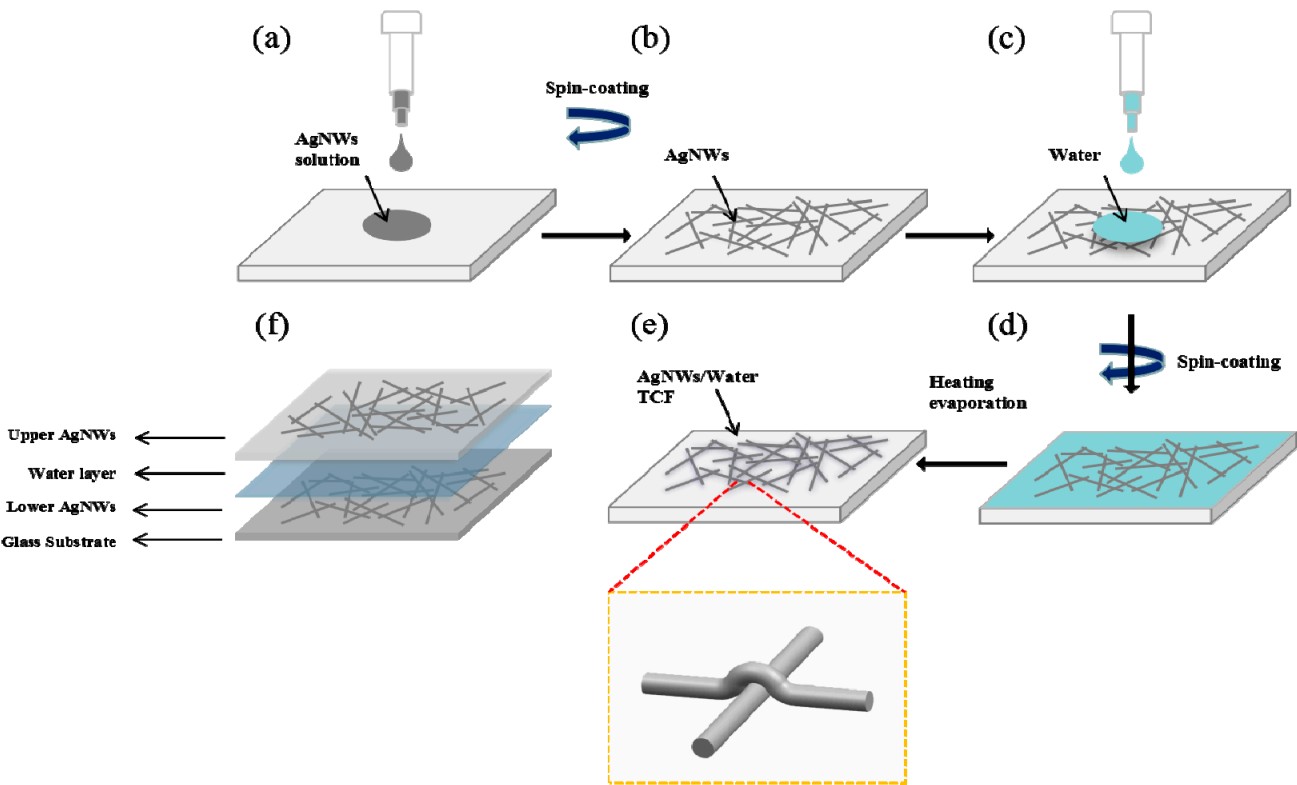

**Figure 1.** AgNWs-Moisture-AgNWs treatment scheme diagram of AgNWs-TCF. (**a**) AgNWss solution was dropped; (**b**) spin-coating deposition; (**c**) deionized water was dropped; (**d**) spin-coating deposition and heating; (**e**) AgNWs/Water TCF after Water moisture treatment; (**f**) layers structure of thin films.

### 2.3. Characterization

A scanning electron microscope (SEM)(JSM-6700F, JEOL Company, Tokyo, Japan) was used to characterize the films' surface morphology. The film surface roughness was characterized by an atomic force microscope (AFM) (Dimension Icon, Bruker Company, Karisruhe, Germany). A light transmittance test was carried out using an Ultraviolet-visible light graduation meter (UV) (Bankman-Du 8B Spectrophono-meter, Pullout General Instrument Company, Beijing, China). A four-probe resistance tester was used for electrical performance measurement (KDY-1, Guangzhou Kunde Company, Guangzhou, China). The bending test used a bending test machine made by the laboratory. The adhesion between silver nanowire film and substrate was tested with 3M adhesive tape.

### 3. Results and Discussion

#### 3.1. Electrical Properties of AgNWss Transparent Conductive Films

The average sheet resistance measured at 25 positions before and after moisture treatment is 53.52 and 20.36 $\Omega$/sq, respectively. The sheet resistance of the AgNWss-TCF after moisture treatment is 62% lower than that before. At the same time, it can be seen from Figure 2 that the uniformity of electrical properties of samples treated with moisture is perceptibly improved. We think that the reason for the high resistance of AgNWss-TCF is that the wire–wire contact at AgNWs network nodes is not tight, and the capillary force generated by moisture treatment can make good ohmic contact at AgNWs network junction. In the process of moisture treatment, water molecules tend to condense and gather at the junctions of silver nanowires with large curvature and fill the gaps between silver nanowires. When the water evaporates, a meniscus-shaped capillary bridge will be formed between silver nanowires, resulting in capillary force during the drying process. According to the hypothesis of the liquid bridge connection model in literature

(Figure 3) [31], the capillary force's value can be estimated analytically by the following formula.

$$F = -\frac{2\pi R \gamma \cos \theta}{1 + (H/2d)} \tag{1}$$

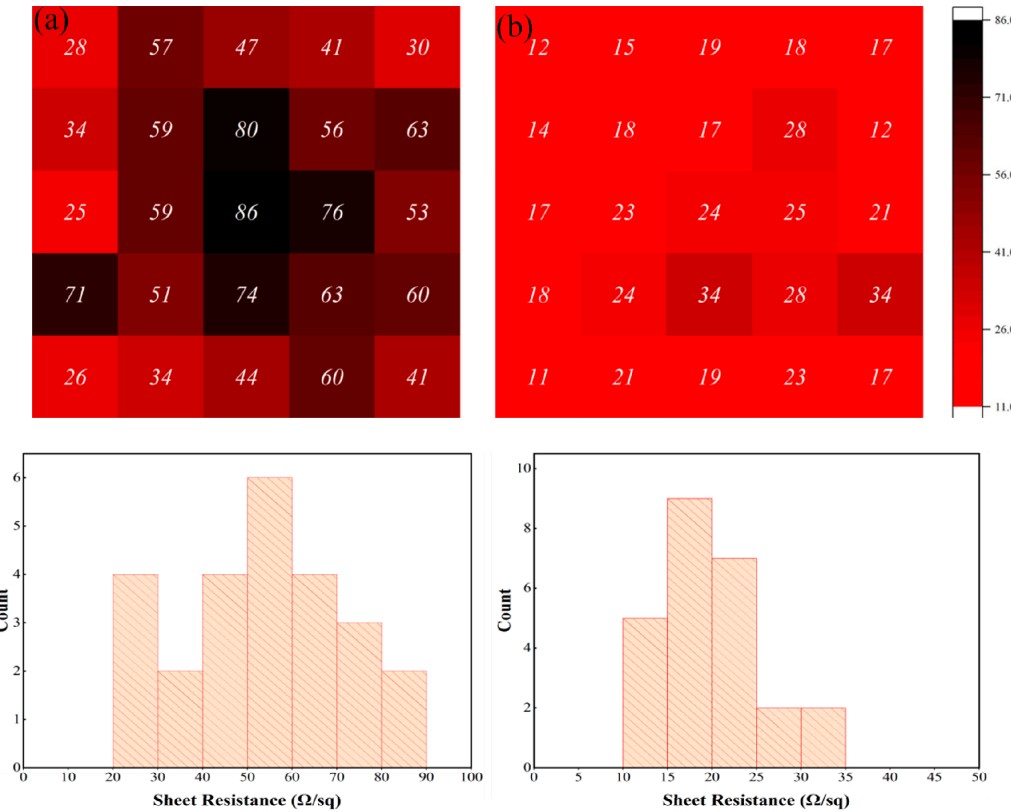

**Figure 2.** Sheet resistance distribution of silver nanowires before and after moisture treatment: (**a**) before moisture treatment of AgNWs-TCF, (**b**) after moisture treatment of AgNWs-TCF.

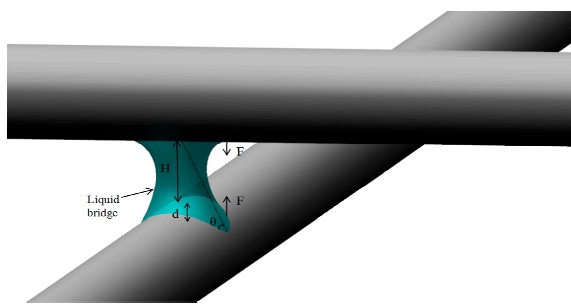

**Figure 3.** Schematic diagram of capillary force.

In which $\gamma$ is the liquid surface tension, $R$ is the diameter of nanowire, $\theta$ is the contact angle, $H$ is the distance between two separated nanowires, and $d$ is the immersion length, which is determined by:

$$d = (H/2) \times \left[ -1 + \sqrt{1 + 2V/\pi R H^2} \right] \tag{2}$$

where $V$ is the liquid volume. Assuming the surface tension of water and the nanowire radius are 71.97 mN·m$^{-1}$ and 50 nm, respectively, when $V = 1 \times 10^3$ nm$^3$, $\theta = 60°$ and $H = 10$ nm, the capillary force is 0.66 nN. When two nanowires are driven closer, the capillary force becomes more extensive. When the two spheres contact the drying process,

the capillary force increases significantly to about 10 nN. The capillary force of two silver nanowires at the junction of AgNWs makes the separated nanowires contact with each other tightly. Theoretically speaking, as the contact area is close to 0, the pressure will reach about 10 MPa when wires–wires in AgNWs network nodes contact each other. The contacted silver nanowires can be fully welded with such a high pressure, as shown in Figure 4b. The self-assembled cold welding of wire–wire junctions can be realized at the AgNWs network nodes, with good ohmic contact and reduced junction resistance.

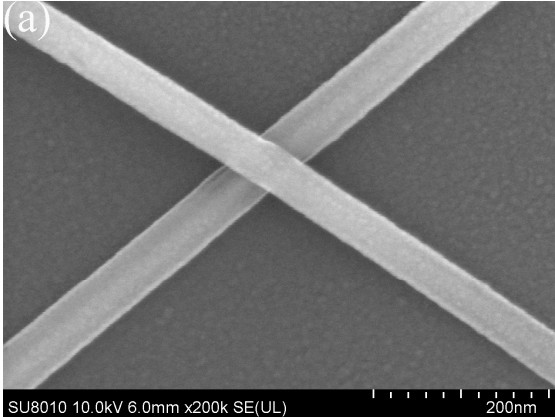 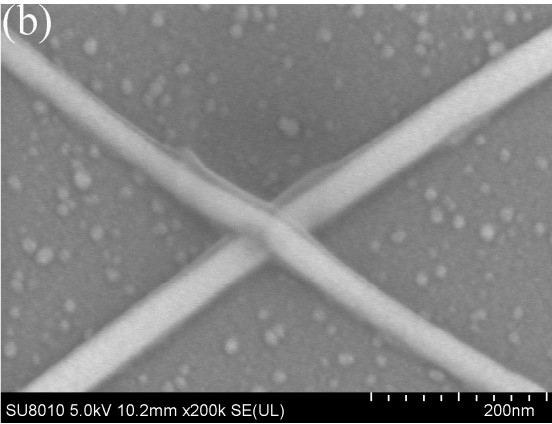

**Figure 4.** SEM images of AgNWs-TCF before and after moisture treatment: (**a**) SEM image of AgNWs-TCF before moisture treatment, (**b**) SEM images of AgNWs-TCF after moisture treatment.

Figure 4 shows SEM images of AgNWs-TCF samples before and after moisture treatment. It can be seen from Figure 4a that silver nanowires of AgNWs-TCF samples are loosely stacked before moisture treatment, and there is a particular gap between silver nanowires at AgNWs network nodes, which is the reason for the high resistance of AgNWs-TCF. It can be seen from Figure 4b that the wires-wires in AgNWs network nodes of AgNWs-TCF sample after moisture treatment are in close contact with each other so that the self-assembled cold welding of wire junctions is realized at the AgNWs network junctions, which has good ohmic contact and reduces junction resistance.

Figure 5 shows AFM images of AgNWs-TCF samples before and after moisture treatment. Figure 5a,c are AFM images of AgNWs-TCF samples before moisture treatment. It can be seen that their maximum and minimum height value is 143.3 and −94.8 nm, respectively, and their differential value is 238.1 nm. As a comparison, Figure 5b,d are AFM images of AgNWs-TCF samples after moisture treatment. It can be seen that their maximum and minimum height value is 125.1 and −76.2 nm, respectively, and their differential value is 201.3 nm. Therefore, AFM images proved that after moisture treatment, the surface roughness became smaller, and the AgNWs network became flat, indicating that the close contact between wires-wires in the AgNWs network improved its electrical uniformity.

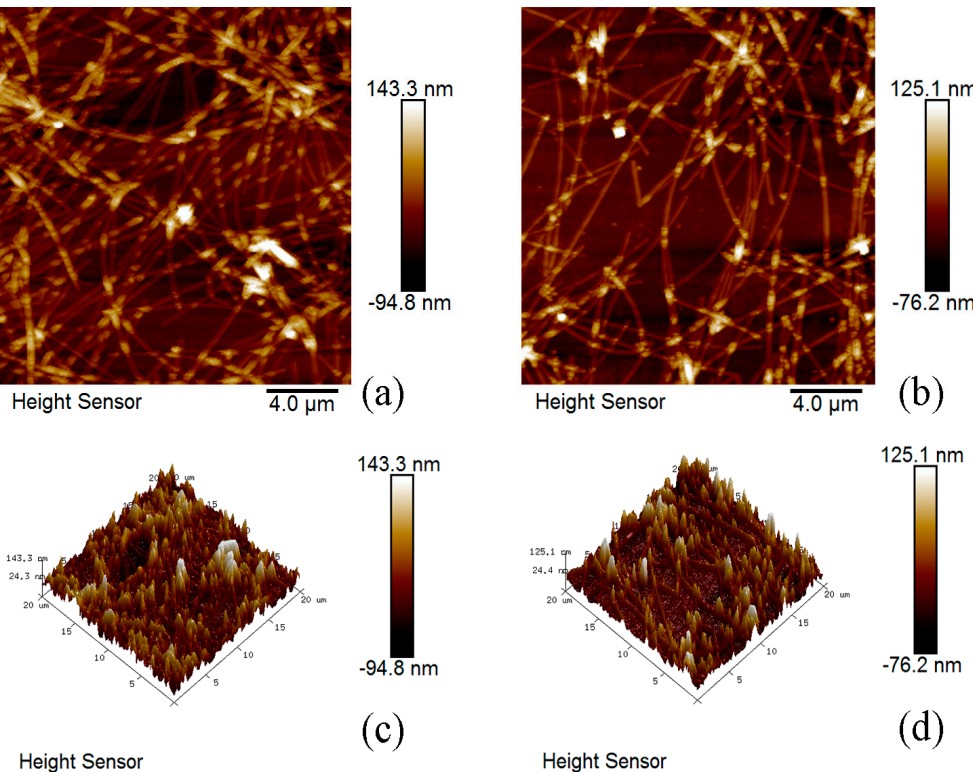

**Figure 5.** AFM images of AgNWs-TCF before and after moisture treatment: (**a**) AFM images of AgNWs-TCF before moisture treatment, (**b**) AFM images of AgNWs-TCF after moisture treatment; (**c**,**d**) correspond to the 3D images in (**a**,**b**), respectively.

### 3.2. Electrical stability of AgNWs Transparent Conductive Films

To study the effect of moisture treatment on the mechanical bending resistance of AgNWs-TCF, we tested the electrical properties of AgNWs-TCF before and after moisture treatment after bending, and the bending radius of each cycle was 1.5 cm. The results are shown in Figure 6. It can be seen from the figure that the sheet resistance of AgNWs-TCF before moisture treatment has increased about three times from 53.52 to 248.50 $\Omega$/sq, after bending 50 times. The sheet resistance of AgNWs-TCF after moisture treatment changed from 20.36 to 81.30 $\Omega$/sq after 50 bends, and the sheet resistance only increased by 60.94 $\Omega$/sq. Therefore, the effect of moisture treatment on improving the mechanical bending resistance of AgNWs-TCF is noticeable. We think that there are two reasons for this result. Firstly, after moisture treatment, the welding between wires–wires at the AgNWs-TCF network junction was realized due to the capillary force discussed above, which increases its mechanical bending resistance. Secondly, the binding force was improved between silver nanowires in AgNWs-TCF and substrates due to capillary force after moisture treatment.

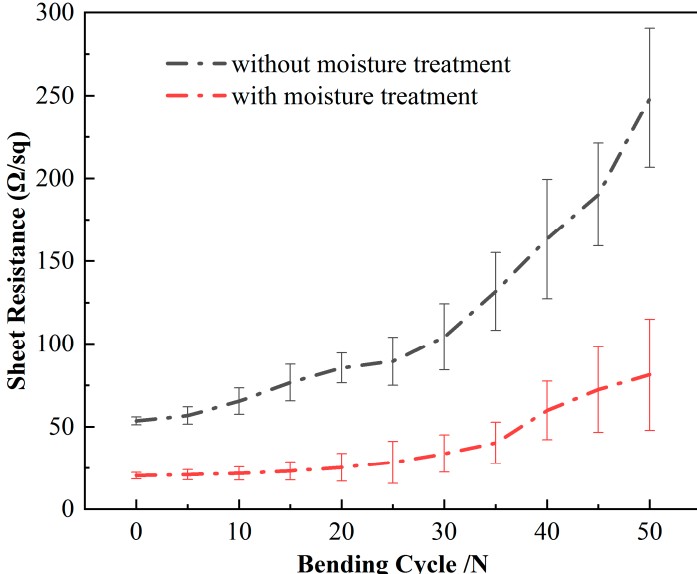

**Figure 6.** Bending test of AgNWs-TCF before and after moisture treatment. The bending radius is 1.5 cm, and it is bent 50 cycles.

In the process of synthesizing AgNWs by reducing $AgNO_3$ with ethylene glycol, PVP is used as the capping agent for the preferential adsorption of silver particles to limit the growth of silver nanowires according to specific crystal planes, which will form a PVP coating on the surface of the silver nanowires. Due to the strong hygroscopicity of PVP, after a small amount of water is introduced into the film surface, PVP softens and deforms in the process of heating and drying at 100 °C for 10 min, forming a nanostructure with larger curvature as shown in Figure 7. Figure 7a is a SEM image of the pristine AgNWs-TCF, and it can be seen that the surface of AgNWs is coated with a smooth PVP layer with a width of 5–6 nm. Figure 7b–f are SEM images of AgNWs films treated with 0.1–0.5 mL water moisture treatment, and it can be seen that the precipitation thickness of PVP increases with the increase of the amount of water. When the amount of moisture treatment is 0.1, 0.2, 0.3, 0.4, and 0.5 mL, the precipitation thickness of PVP is 7–8 nm, 10–13 nm, 14–16 nm, 17–19 nm, and 21–24 nm, respectively. This rough PVP layer is conducive to forming a curved nano-space structure with the substrate, generating capillary force, enhancing the bonding between silver nanowires and the substrate, thus improving the bending resistance of AgNWs-TCF.

Figure 8 shows the combination of silver nanowires and substrate before and after moisture treatment. In the experiment, 3M adhesive tape was used to peel the AgNWs-TCF surface six times. Figure 8a is a SEM image of AgNWs-TCF before moisture treatment. It can be seen that most silver nanowires on the substrate surface have been removed after repeated adhesive tearing six times, and four probes can no longer measure the resistance value. Figure 8b is a SEM image of AgNWs-TCF after moisture treatment. It can be seen that the silver nanowires are torn off to a much lesser extent on the AgNWs-TCF surface than that of the sample before moisture treatment after repeated adhesive tearing six times, and the resistance value measured by four probes is 218 Ω/sq. This indicates that moisture treatment can enhance the bonding between silver nanowires and substrate, which is also why moisture treatment can improve the mechanical bending resistance of AgNWs-TCF.

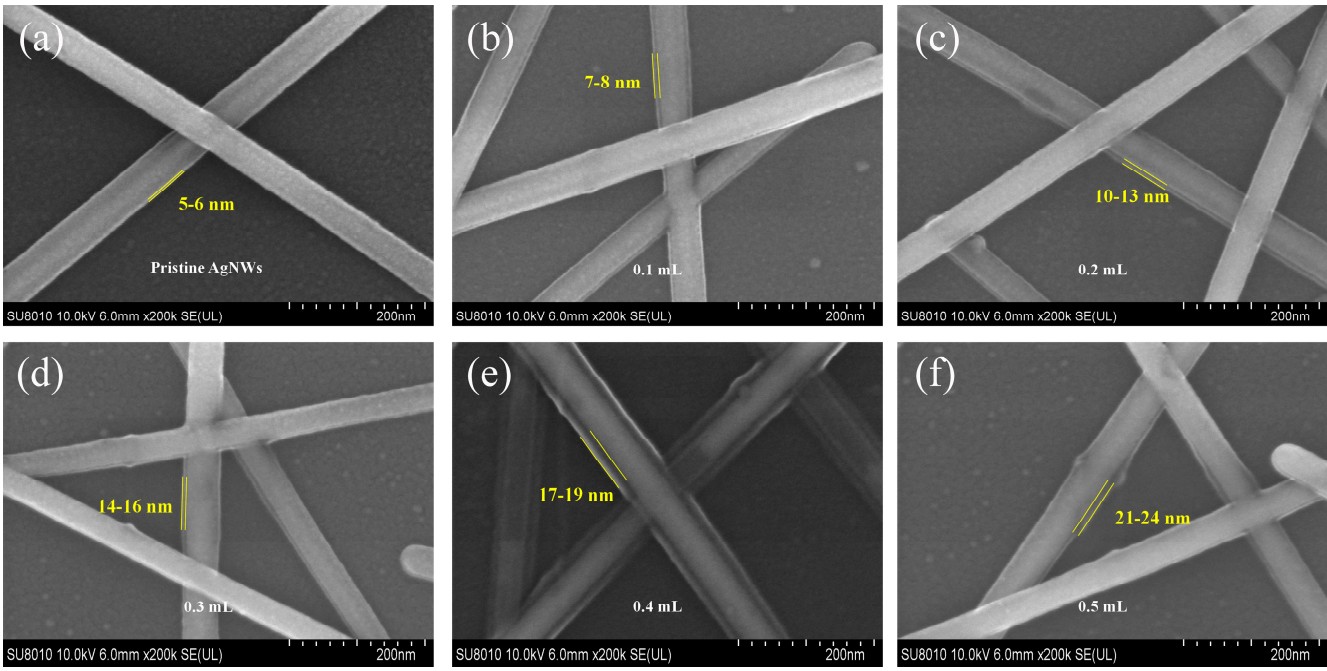

**Figure 7.** SEM images of PVP wrapping on the surface of AgNWss: (**a**) pristine AgNWss; (**b**–**f**) PVP precipitation thickness on the surface of AgNWss treated with different water moisture amounts for (**b**) 0.1 mL, (**c**) 0.2 mL, (**d**) 0.3 mL, (**e**) 0.4 mL and (**f**) 0.5 mL.

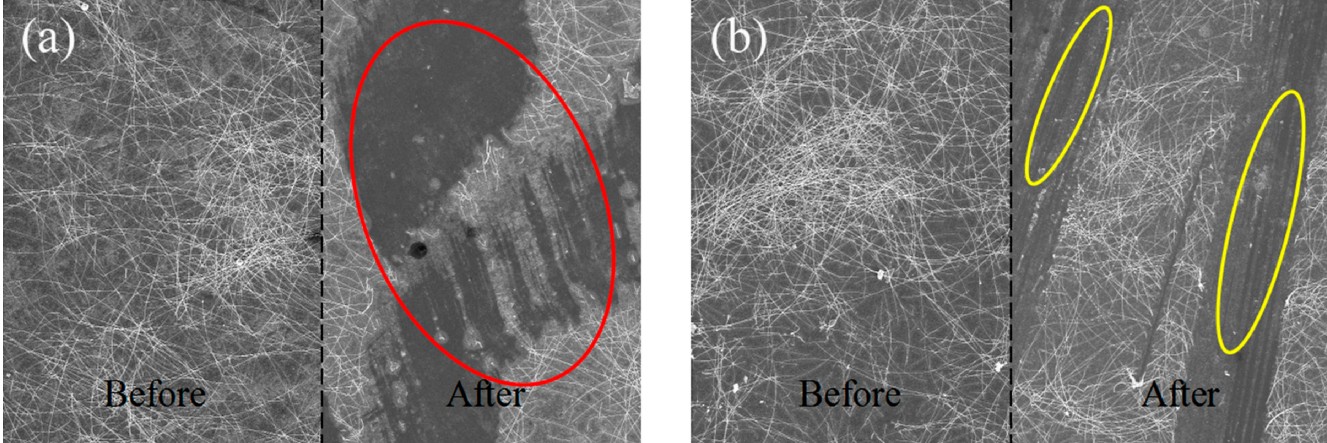

**Figure 8.** SEM pictures of adhesive test between AgNWs and substrate before and after moisture treatment: (**a**) SEM pictures of AgNWs-TCF surface before moisture treatment, (**b**) SEM pictures of AgNWs-TCF surface after moisture treatment.

Figure 9 shows the environmental stability results of the electrical properties of AgNWs-TCF before and after moisture treatment. It can be seen from the figure that the sheet resistance of AgNWs-TCF before moisture treatment increased from 53.52 to 86.71 Ω/sq with a increasing amplitude of 62.07% after exposure in the air environment for 28 days, but the sheet resistance of AgNWs-TCF after moisture treatment increased from 20.36 to 32.44 Ω/sq, showing evident electrical performance and environmental stability. The improvement of environmental stability of AgNWs-TCF electrical properties by moisture treatment may be related to the wire–wire junction welding at silver nanowire mesh and the close combination of silver nanowire and substrate. On the one hand, after moisture treatment, the PVP protective film wrapped on the surface of the silver nanowire becomes thicker, as shown in Figure 7, and its thickness increases from 5 nm before moisture treatment to 21 nm after moisture treatment. The thickening of PVP layer effectively

protects the silver nanowires from oxidation and corrosion of water vapor and acid in the air. In contrast, as reported by Ye et al., Zn, Sn, and other metals were used to overcome the oxidation problem of CuNWs [32,33], and water moisture treatment not only improved the electrical conductivity of the film, but also solved the problem of nanowires oxidation. On the other hand, after moisture treatment, the AgNWs network node and AgNWs are tightly combined with the substrate, significantly reducing the corrosion area of silver nanowires by water vapor and acidic substances in the air.

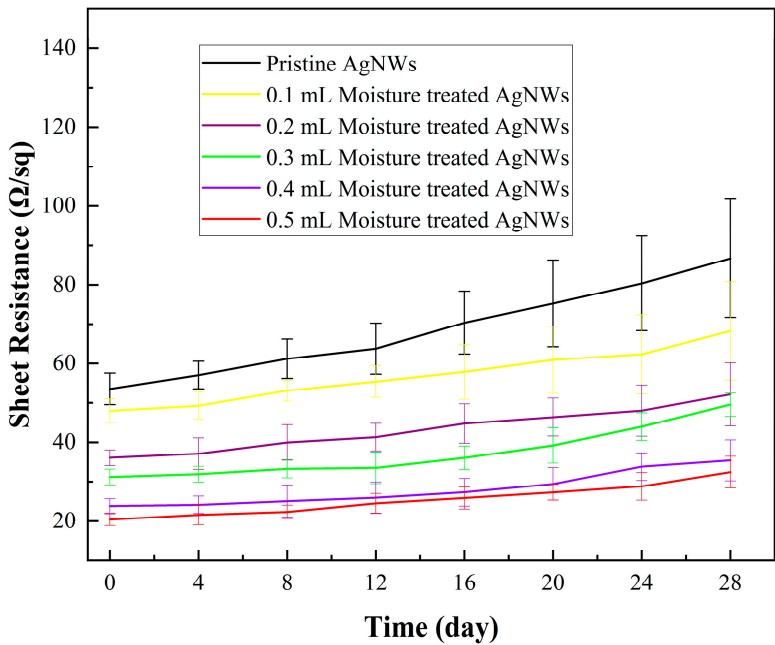

**Figure 9.** Test results of electrical stability in air environment of AgNWss-TCF before and after moisture treatment.

### 3.3. Optical Properties of AgNWs Transparent Conductive Films

In order to explore the influence of water moisture amount on the light transmittance of AgNWs thin films, we measured the optical transmittance of AgNWs-TCF in the visible wavelength range with the original AgNWs films and water moisture amount of 0.1, 0.2, 0.3, 0.4 and 0.5 mL, respectively. As shown in Figure 10, it is obvious that the light transmittance increases with the increase of water mist amount. The light transmittance at the wavelength of 550 nm is 82.5%, 83.9%, 84.6%, 85.1%, 86.9% and 87.3%, respectively. This is due to the slight movement of nanowires caused by water moisture treatment, which increases the tightness of conductive network nodes, thus reducing the refractive index of light between nanowires.

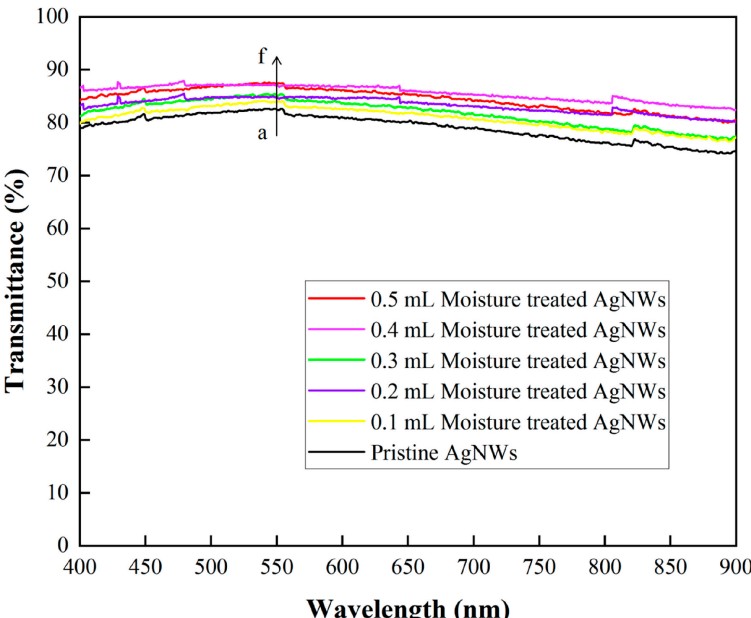

**Figure 10.** Increase of water moisture quantity and light transmission performance in visible light range.

## 4. Conclusions

In this paper, an AgNWs-Moisture-AgNWs treatment scheme of AgNWs-TCF was introduced. The results show that the AgNWs thin film treated by moisture could realize the cold welding of wire-wire junctions at AgNWs network nodes, and the AgNWss are closely bonded with the substrate. It is beneficial to improve the electrical properties of AgNWs-TCF and improve the uniformity of its electrical properties. Simultaneously, moisture treatment for AgNWs-TCF can improve its electrical stability in the air environment, mechanical resistance to bending, and resistance to friction of AgNWs-TCF surface. This method does not need additional instruments and equipment, and it is simple to operate and has commercial potential.

**Author Contributions:** Conceptualization, Y.H. and L.Z. (Lipeng Zhou); Formal analysis, Y.H.; Investigation, Y.G., H.G., and W.Z.; Methodology, Y.H., L.Z. (Lipeng Zhou), L.Z. (Lilin Zhan), H.L., and Y.C.; data curation, Y.H., L.Z. (Lipeng Zhou), P.W., D.W., F.H., and K.Z.; Supervision, Y.H., W.L., and N.C.; Writing—original draft, Y.H. and L.Z. (Lipeng Zhou); Writing—review and editing, Y.H., L.Z. (Lipeng Zhou), and K.H. All authors have read and agreed to the published version of the manuscript.

**Funding:** This research was funded by the National Natural Science Foundation of China (Grant No. 62041405 and 51802131), the Natural Science Foundation of Jiangxi Province, China (Grant No. 20192BAB217012, 20202BAB202011 and 20202BABL214011), the Key R & D Program of Jiangxi Province, China (Grant No. 20192BBE50056 and 20171BBE50053), the Postgraduate Innovation Special Fund Project of Jiangxi Province, China (Grant No. YC2020-S422).

**Institutional Review Board Statement:** Not applicable.

**Informed Consent Statement:** Not applicable.

**Data Availability Statement:** The data presented in this study are available on request from the corresponding author.

**Conflicts of Interest:** The authors declare no conflict of interest.

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
