# Peer review of "Moisture-Assisted Formation of High-Quality Silver Nanowire Transparent Conductive Films with Low Junction Resistance"

_coatings, doi:10.3390/coatings11060671_

Round 1

Reviewer 1 Report

The paper presents a very interesting topic dealing with Silver nanowires transparent conductive oxide which is currently being one of very interesting materials that may replace ITO soon acting as a the next-generation transparent conductive film. Till now, the biggest setback of this material was its liability to be easily oxidated and the high wire-wire junction resistance. Both of these setbacks dramatically limit the commercial application of this promising material. 

Authors of this paper suggested a breakthrough approach based on moisture treatment so that the surface was covered with a layer of water vapor making the morphology of silver nanowire rather complex, and the curvature large. The authors also proved that AgNWs-TCO treated by moisture has significantly better conductivity, with an average sheet resistance of 20 Ω/sq and more uniform electrical properties, which I personally consider to be one of the biggest benefits of this work. 

The results presented in this paper are additionaly supported by the necessary bending test and adhesion test showing that AgNWs-TCO treated by moisture still exhibited good mechanical bending resistance and environmental stability.

In my opinion, the paper clearly brings novel findings by adapting a new approach that may prove itself to be very interesting for preparation of Silver nanowires transparent conductive oxide that may later easily find number of applications both in vacuum and in special atmospheric conditions.

I congratulate to authors and suggest the paper to be accepted in the present form.

Author Response

Please find the author reply in attachment.

Reviewer 2 Report

The authors investigated various properties of transparent conductive films based on silver nanowires and the modification of their properties after moistening and drying. The topic of the article is relevant, since there is an active study of transparent conductive films for various applications. There are many experimental results in this article that deserve publication.
However, the article contains some shortcomings that it is desirable to eliminate before publication.

1. The term AgNWs-TCO is not clear. It would be more correct to say that it is a transparent conductive film (TCF), but not an oxide (TCO).
2. Unfortunately, there are shortcomings in the description of materials and methods. Nothing was said about the composition of the AgNW solution, the characteristics of nanowires. It is also not clear what role PVP plays in the process. Its concentration and method of application are also not disclosed.
3. Figure 1 shows the sequence of the AgNWs-Moisture-AgNWs process, however, the "before" and "after mousture treatment" samples are compared. It is not clear whether samples with one or two layers of AgNWs or the same sample with additional water treatment.
4. The English language also deserves a thorough revision. In particular, the title of the article, it seems to me, is formulated incorrectly.

Only if all these questions are clarified, it will be possible to assess the significance of the article.

Author Response

(The authors gave the same response as above.)

Reviewer 3 Report

This paper presents the development of a simple and cost-effective moisture treatment method to realize the tight connection between wire-wire in AgNWs network nodes and produce good ohmic contact between them. Considering the intense interest in silver nanowire based transparent conductor as an alternative to ITO, this paper may provide a useful method. And this may be recommended for publication if the authors can respond to the following revision comments,

1) During the sample preparation, the wet film of AgNWs was heated and dried at 100 ℃ for 10 min to evaporate solvent residue, and AgNWs-TCO was obtained after cooling to room temperature. Then, a small amount of deionized water dripped on the surface of AgNWs-TCO and those process was repeated. The authors discussed that the moisture treatment enhanced the connection between nanowires. However, between each NW deposition step, there was 100C heating for 10 minutes to remove the residue solvent, but this heating can be enough to induce the thermal annealing effect to reduce the contact resistance between nanowires. I wonder how the authors could remove this thermal annealing effect from the moisture treatment effect.

2) Besides DI water, what is the requirement for the liquid material for successful process development?

3) Even though the authors used different solvent (DI water) to enhance the contact between silver nanowires, similar concept has been also demonstrated for other materials such as conducting polymer (PEDOT:PSS) ink assisted Joining (Adv. Funct. Mater., 23, 4171 (2013)). The difference of those previous research should be discussed in the introduction part.

4) The bad connection between wire-wire in AgNWs network nodes is one of the main problem in Ag NW transparent conductor. The authors briefly listed the current technologies to enhance the junction resistance such as metal oxide coating (J. Mater. Chem. C 2014, 2, (19), 3750-3755), and graphene material modification (Applied Sciences 302 2020, 10, (14)), ultraviolet lamp irradiation (Nat Mater 2012, 11, (3), 241-9.), thermal annealing (Nano Research 2018, 11, (4), 1998-2011.) to solve the problem of high wire-wire junction resistance of AgNWs network. However, there are more practical advances such as laser nanowelding (Adv. Mater., 24, 3326 (2012); Nanoscale, 4, 6408 (2012)), nano-joining with CNT (Adv. Funct. Mater., 24, 5671 (2014)) and applying electrical current (Nano-Micro Letters, 6, 293 (2014)). Those recent progress also needs to be discussed in the introduction part.

5) The authors mentioned ‘However, due to the disadvantages of AgNWs, such as easy oxidation and high wire-wire junction resistance, which dramatically limits its commercial application. However, Ag NW has little oxidation problem even though it slightly oxidizes. Among metal nanowires, copper nanowire shows severe oxidation problem (Acc. Chem. Res. 2016, 49, 3, 442–451) and there are many new methods (Adv. Mater., 26, 5808 (2014); Adv. Mater. Tech., 2000661 (2020)) to overcome the oxidation problem in copper nanowire as transparent conductor. The oxidation issue in Ag NW needs to be corrected and the competing metal nanowire (copper nanowire) should be discussed in the manuscript.

Author Response

(The authors gave the same response as above.)

Round 2

Reviewer 3 Report

The authors responded well to the comments.